# Association between Problematic Use of Smartphones and Mental Health in the Middle East and North Africa (MENA) Region: A Systematic Review

**DOI:** 10.3390/ijerph20042891

**Published:** 2023-02-07

**Authors:** Samira Bouazza, Samira Abbouyi, Soukaina El Kinany, Karima El Rhazi, Btissame Zarrouq

**Affiliations:** 1Laboratory of Epidemiology and Research in Health Sciences, Faculty of Medicine and Pharmacy, Sidi Mohamed Ben Abdellah University, Fez 30070, Morocco; 2Faculty of Sciences and Techniques, Errachidia, Moulay Ismail University of Meknes, Errachidia 52000, Morocco; 3Department of Biology and Geology, Teachers Training College (Ecole Normale Superieure), Sidi Mohamed Ben Abdellah University, Fez 30030, Morocco

**Keywords:** problematic smartphone use, depression, anxiety, stress, MENA region

## Abstract

Smartphones have become essential components of daily life, and research into the harmful effects of problematic smartphone use (PSU) on mental health is expanding in the Middle East and North Africa (MENA) region. This issue has yet to be synthesized and critically evaluated. To find quantitative observational studies on the relationship between PSU and mental health in the MENA region, we developed a search equation and adapted it for four databases. The Preferred Reporting Items for Systematic Review and Meta-Analysis (PRISMA) guidelines were followed during the selection process. This review included 32 cross-sectional studies and one cohort study. The available language was English. All identified studies published until 8 October 2021, were considered. A modified Newcastle-Ottawa scale was used to assess the quality of the included studies. The studies enrolled 21,487 people and had low-to-moderate methodological quality. The prevalence of PSU ranged from 4.3 to 97.8 percent. The time factor, type of application used on the smartphone, and sociodemographic characteristics were the determinants of PSU. Depression, anxiety, and stress were strongly correlated with PSU. Epidemiological longitudinal studies that respect the quality of evidence are needed in all MENA countries to better plan and implement preventive measures against PSU.

## 1. Introduction

Smartphones have rapidly evolved, and the diverse tasks they can perform have led to their use in a wide range of daily activities. We are compelled to regularly use our smartphones for answering calls, responding to messages, searching for information, working/studying online, and distracting ourselves [1]. Checking smartphones anywhere, anytime, and under any circumstances has become a habit that can evolve into problematic behavior when associated with unexpected outcomes [2].

Several studies have found that smartphone use is associated with numerous dysfunctions in daily life. These flaws affect physical and mental health, social relationships, and academic as well as professional achievement [3,4,5]. Problematic smartphone use (PSU) has been linked to physical health issues, such as neck and wrist pain, eye discomfort, and sleep disorders [3,4,5], as well as mental disorders, such as depression, anxiety, and stress [3,4,5,6,7].

The concept of smartphone addiction (dependency) originated in the wake of a series of studies examining the harmful impacts of smartphones on users’ daily functioning, albeit it has yet to be fully characterized [2,5]. In 2012, Billieux defined problematic cellphone use as “an inability to regulate one’s use of the mobile phone, which eventually involves negative consequences in daily life” [8] (p. 1). In 2016, De-Sola et al. argued that there is considerable comparability between PSU and substance use disorder [9] because many of the symptoms of PSU, such as withdrawal, tolerance, craving, salience, and lack of control, are also symptoms of substance use disorder [4,7]. Other surveys, on the other hand, have overlooked the occurrence of withdrawal, tolerance, and loss of control symptoms in PSU [10]. The disparity between the findings of studies in this field, combined with the poor number of neurobiological studies that have invested in this area, resulted in a lack of agreement on how to define PSU and its subsequent recognition as a type of behavioral addiction by the 5th edition of the Diagnostic and Statistical Manual of Mental Disorders (DSM5). Gambling disorder is indeed the only recognized behavior as a non-substance-related addictive disorder. Other behaviors that have been studied for their addictive potentials, such as hypersexuality, shopping, and internet use, have not been admitted to this section for the lack of sufficient evidence [1,11,12].

Despite the disagreement over PSU definition, the number of studies investing in this field is growing and exploring various phenomena that concurrently manifest with excessive smartphone use. These studies continue to reveal the harmful effects of PSU, which piques the interest of public health officials around the world. As a result, numerous psychometric scales have been developed to identify individuals suffering from PSU [10]. These scales have been developed in several languages, including English (Problematic Mobile Phone Use Questionnaire-Revised: PMPUQ-R), Chinese (Mobile Phone Addiction Index: MPAI), Turkish (Mobile Addiction Scale: MAS), Korean (Smartphone Addiction Scale: SAS, Smartphone Addiction Scale Short Version: SAS-SV), Spanish (Mobile Phone Addiction Craving Scale: MPACS), and Arabic (untitled smartphone addiction scale) [10].

At the same time, several examinations of the association between PSU and mental health distress have been carried out, and they have confirmed the strong relationship between these two issues [3]. Among the determinants of PSU, there are those which are linked to the sociodemographic characteristics of individuals [3]; we tried to focus on the population of the Middle East and North Africa (MENA) region, which share several socio-cultural traits. The goal behind this is to systematically review and synthesize evidence on the prevalence of PSU and its relationship with mental health disorders. Determining the relationship between PSU and mental health has important implications for raising public awareness about the impact of PSU on mental health and also for the development of clinical guidelines adapted to the socio-cultural context of this population to minimize or prevent mental health illness among smartphone users of the MENA region.

A systematic review of studies that were conducted in the MENA region to reach this objective, which focused on the association between PSU and mental health problems, mainly depression, anxiety, and stress, was carried out.

Various criteria were taken into consideration in the selection of the MENA region. First, except for Iran, all countries in the MENA region have Arabic as their official language [13] and subsequently share some traditions and customs. Second, this region is experiencing rapid demographic growth. In 2020, for instance, the population of this region represented 5.4% of the world population, and young people under 35 years old, who may be vulnerable to PSU, account for two-thirds of the MENA population [14]. Third, according to the International Telecommunication Union (ITU), smartphone ownership in Arab countries reached more than 80% in 2017, with increasing internet penetration in this population’s households [15]. Statistical data for the year 2022 indicates that the internet penetration rate was 84.1% in Morocco, 60.6% in Algeria, 71.9% in Egypt, and 98% in Qatar, while the mobile connection rate was 129.3% in Morocco, 103.5% in Algeria and 93.4% in Egypt [16]. As a result of the problematic use of smartphones, the MENA population is becoming increasingly vulnerable to the risks of mental illness, endangering their well-being and quality of life. Moreover, the public health system in this region suffers from insufficient mental health expenditure and interventions [17]. Given that this review is part of a research project focusing on the impact of PSU on the mental health of young Moroccans and that social risk factors for PSU exist, it is critical to have data on this subject from other countries that are socio-culturally close, such as those in the MENA region.

## 2. Materials and Methods

### 2.1. Search Strategy

The PRISMA statement guidelines were followed in the conduct of this systematic review [18]. Additionally, an a priori protocol was registered on PROSPERO under the number CRD42022266732. All quantitative observational studies in the MENA region that looked at the relationship between PSU and mental health were found in four databases: PubMed, Web of Science, ScienceDirect, and Cochrane. The only language available was English. References to included articles were individually checked to ensure that no articles were overlooked. All identified papers published up to 8 October 2021 were taken into account.

### 2.2. Search Terms

A variety of terms were employed to ensure the detection of all studies concerning the PSU and its relationship to mental health in the MENA region. The search equation that was devised was as follows: (“smartphone addiction” OR “smart phone addiction” OR “smartphone problematic use” OR “smartphone overuse” OR “smartphone abuse” OR “smartphone use” OR “mobile phone dependence” OR “mobile phone addiction” OR “mobile phone problematic use” OR “mobile phone overuse” OR “ mobile phone use” OR “cellular phone dependence”) AND (mental health OR mental disorder OR psychopathology OR depression OR anxiety OR stress) AND (Morocco OR Moroccan OR Algeria OR Algerian OR Tunisia OR Tunisian OR Libya OR Libyan OR Mauritania OR Mauritanian OR Lebanon OR Lebanese OR Syria OR Syrian OR Jordan OR Jordanian OR Iraq OR Iraqi OR Iran OR Iranian OR Persian OR Israel OR Israeli OR Palestine OR Palestinian OR Sudan OR Sudanese OR Djibouti OR Djiboutian OR Ethiopia OR Ethiopian OR Egypt OR Egyptian OR Qatar OR Qatari OR Bahrain OR Bahraini OR Kuwait OR Kuwaiti OR Oman OR Omani OR united Arab emirates OR Saudi Arabia OR Yemen OR Yemenite OR MENA region OR “middle east” OR “north Africa” OR “Arab world”). This equation has been divided into several parts when searching in the ScienceDirect and Web of Science databases.

### 2.3. Inclusion/Exclusion Criteria

We only included quantitative observational studies (cross-sectional, case-control, and cohort studies) on smartphone use and its association with mental health, specifically anxiety, depression, and stress, in the MENA region while excluding qualitative and experimental studies from this review.

### 2.4. Data Extraction

Two authors independently reviewed all identified studies for the relevance of the inclusion/exclusion criteria. They extracted specific data from each study, including the first author’s name, year of study, study design, country, sample size, sociodemographic characteristics of the study population (age and gender), PSU definition and prevalence, and PSU and mental health assessment tools.

### 2.5. Quality Assessment

A modified Newcastle-Ottawa scale was used to assess the methodological quality of each study included in this systematic review [19,20]. This scale concentrated on three study domains: selection, comparability, and exposure. Studies in the selection domain can be classified as poor quality (0 or 1 star), moderate (2 stars), or good quality (3 stars). In the domain of comparability, studies can be sorted as poor quality (0 stars) or good quality (1 or 2 stars). While for the third domain, exposure, studies can be categorized as poor quality (0 or 1 star), moderate quality (2 stars), or good quality (3 stars). The GRADE system was used to assess the quality of evidence in the included studies [21].

## 3. Results

The search yielded 14,280 references. After excluding duplicate studies from PubMed, Cochrane, Web of Science, and ScienceDirect, and after stepwise exclusion of research outside the scope of our review, 29 articles remained for inclusion in this systematic review. We discovered four additional relevant papers by consulting the references of the included studies, which resulted in the inclusion of 33 original studies published between 2014 and 2021 in our systematic review. Figure 1 depicts a flowchart of the research strategy and study selection process.

One of the thirty-three articles included for review was a cohort study, and the other thirty-two were cross-sectional studies. Iran had the most studies (n = 12), followed by Lebanon (n = 6), Saudi Arabia (n = 4), United Arab Emirates (n = 3), Egypt and Israel (n = 2 each), and one study each from Kuwait, Jordan, and Oman. There were no studies that met our eligibility criteria from Algeria, Bahrain, Djibouti, Ethiopia, Iraq, Mauritania, Morocco, Libya, Palestine, Qatar, Sudan, Syria, Tunisia, or Yemen (Table 1).

Table 2 and Table 3 shows the quality ratings of the cross-sectional and cohort studies that were chosen (Table 2 and Table 3). Three domains were consulted to define the quality assessment of included studies. The first was the exposure domain. All included studies relied on self-report, and the statistical test used to analyze data was clearly described, as was the measurement of the association, which involved the probability level and/or confidence interval. Subsequently, all the included studies were classified as moderate in this exposure domain. The selection domain was the second domain considered in this classification, with 26 studies of high quality according to this domain [22,23,24,25,26,27,28,29,30,31,32,33,35,38,39,41,43,44,45,46,48,50,51,52,53,54]. Comparability was the third domain. In this domain, the fifteen studies that controlled for confounding factors while analyzing the associated variables with PSU were of high quality [22,25,27,29,31,32,34,38,41,42,43,44,48,51,52]. The modified Newcastle-Ottawa scale and Table 4 were used to assess the overall quality of the studies [19,20] (Table 4). The current systematic review included studies of moderate (15 studies) or poor (18 studies) methodological quality.

### 3.1. PSU Definition

There were several definitions of problematic smartphone use. According to some authors, it is the inability to limit phone use that eventually disrupts the users’ functioning [23,24,25,44,45,49], while other authors claimed that problematic smartphone use was a type of behavioral addiction that included salience, preoccupation, tolerance, withdrawal, and compulsive symptoms [27,28,32,43,49,50,52].

### 3.2. PSU Measurement Tools

A total of 13 validated scales were used to assess the prevalence of PSU. Smartphone Addiction Scale Short Version (SAS-SV) was the most commonly used scale (used in 12 studies), followed by Smartphone Addiction Scale SAS (used in 5 studies), while the Mobile Phone Problematic Use Scale MPPUS-10, the Smartphone Addiction Inventory SPAI, the Smartphone Application-Based Addiction Scale SABAS, and the Mobile Phone Addiction Index MPAI were all used in two different papers. The comparison of prevalence values measured by these scales revealed a high degree of variability. The prevalence of PSU, which was measured by SAS-SV in seven studies conducted in Iran, Lebanon, Kuwait, Egypt, and Saudi Arabia [23,26,32,43,49,50,52], varied between 17% and 71.96%, whereas the one measured by SAS was 95.8% in Egypt [35].

### 3.3. Determinants of PSU

PSU was associated with the time factor, variety of smartphone usage, and sociodemographic characteristics, according to the studies included in the current systematic review.

#### 3.3.1. Time Factor

In total, ten studies found that PSU was positively and significantly associated with the average duration of daily mobile use [26,31,35,37,40,42,48,49,50], which was determined as more than 4 h [26,31] or 5 h [48] in the studies conducted by Buabbas A.J. et al., EL-Sayed Desouky et al., and Matar Boumosleh J. et al. Despite this, only a single study considered that the daily time spent using a mobile phone was a significant predictor of PSU [42]. Some studies have highlighted other time modalities affecting smartphone use, such as checking smartphones at night [36,44], higher frequency of mobile phone use during the day, the shorter time until first mobile phone usage in the morning [29,49], the number of hours per session using smartphone [26], and younger age of owning or first use of the smartphone, which was also significantly correlated with PSU [31,48,50].

#### 3.3.2. Variety of Smartphone Usage

The most stated reasons for smartphone use in the studies included in this review were entertainment, calling, and texting [40,48,50]. The majority of features used by individuals were social networking sites and messaging functions [29,37,38,49,50]. Therefore, the PSU tended to be higher when smartphones were used for social media, chatting, or gaming [46,48,49,50].

#### 3.3.3. Sociodemographic Characteristics

The findings of the identified studies on the relationship between gender and PSU were inconclusive. Most studies discovered no significant gender differences in PSU; however, some other studies found that the risk of smartphone addiction was higher in females than males [26,28,31,37,44] and that being female predicts PSU [31,42]. In contrast, other studies have found that males have a higher PSU than females [32,35,40,49].

Studies on the relationship between age and PSU were also inconclusive. While five studies found no link between age and PSU [33,38,42,50,53], other studies found no association between PSU and younger age [27,32,39,41,43,46,48] and that younger age was a significant predictor of PSU [41]. In contrast, a single study claimed the existence of an association between older age and PSU [31]. Regarding marital status, there was no significant association with PSU in seven of the studies examined in this review [25,27,30,41,42,50,53], whereas other studies revealed that PSU was higher in singles than married people [31,32,39,46] and that being single predicts PSU [31].

Other sociodemographic characteristics and PSU, such as income, type of residence, and educational level, have yielded mixed results in epidemiological studies. Concerning the association between PSU and education level, four studies reported that PSU was significantly associated with education level [31,32,48,53]; nine other studies found no relationship between PSU and education level [25,27,29,30,41,42,43,44,50].

### 3.4. PSU and Mental Health

Twenty-three different validated scales and one independent questionnaire were used to assess mental health. Almost all the studies included in this systematic review (33 papers) revealed a decline in mental health with PSU.

#### 3.4.1. Depression

Researchers found a significant positive correlation between PSU and depression in 14,103 participants across 16 studies [22,23,24,25,26,27,28,31,33,35,37,41,42,43,45,50], with magnitudes ranging from 0.164 to 0.996 and significance ranging from 0.001 to 0.01. The prevalence and severity of depression in the MENA region’s population were assessed using a variety of instruments. As a result, according to the Center for Epidemiological Studies Depression Scale (CEDSD-10), the prevalence of depression was 68.6 percent in Emirati students aged 18 to 33 years [42], and 32.7 percent and 50.72 percent in Saudi Arabian and Egyptian students aged 18 to 26 years, respectively [23,31]. The Hamilton Rating Scale of Depression (HRSD) revealed that 8.4 percent of nursing Egyptian students had moderate-to-severe depression [35], and the Patient Health Questionnaire (PHQ-9) in Middle Eastern postgraduate students revealed that 34.1 percent of high smartphone users had moderate-to-severe depression [27].

As for the determinants of depression, the findings of the various studies included in the current systematic review were inconclusive. Some studies found that depression was significantly and negatively correlated with age and that it primarily affected females [26,42,45], whereas other studies found no significant correlation between depression and sociodemographic factors [50].

Some studies found that depression was significantly associated with the duration of smartphone use (r = 0.15; *p* < 0.001) [37], daily smartphone use (r = 0.11; *p* < 0.05) [42], and using smartphones for entertainment [50].

Other mental health problems were positively and significantly correlated with depression according to two studies, which were anxiety (r = 0.6; *p* < 0.001) [37], stress (r = 0.71; *p* < 0.001) [37], rumination (r = 0.38; *p* < 0.01) [28], and fear of missing out (r = 0.43; *p* < 0.01) [28]. However, two studies found that a significant negative correlation exists between depression and self-esteem [35,42] with magnitudes of r = −0.48 (*p* < 0.01) and r = −0.92 (*p* = 0.006), respectively. Self-esteem was measured in these two studies using two various scales (Rosenberg Self-Esteem Scale RSES and Self-esteem Inventory).

#### 3.4.2. Anxiety

The relationship between anxiety and PSU was studied in 13 studies [22,23,25,26,28,30,31,34,37,41,45,48,51], 10 of which [22,23,25,26,28,30,31,34,37,45] found a significant positive correlation between PSU and anxiety in 11,030 participants, with magnitudes ranging from 0.12 to 0.562 and significance between 0.001 and 0.05. For example, in Lebanese students with an average age of 20.64 ± 1.88, it was discovered that each increase of one unit in anxiety scores increases the total score of PSU by 1.7 folds [48], while anxiety is one of the most significant predictors of PSU in Iranian students were aged on average 22.29 ± 3.5 [41]. In contrast, social anxiety explained 31.5 percent of the variance in PSU rating in Israeli students [34].

For the sociodemographic determinants of anxiety, contradictory results were found concerning gender. Thus, female students from Saudi Arabia, United Arab Emirates, and Kuwait had a significantly higher prevalence of trait anxiety than males [26,28,31]; contrariwise, Omani male students were more anxious than females [30].

Concerning mental health problems, they were positively correlated with anxiety, insomnia (r = 0.439; *p* < 0.05), stress (r = 0.895; *p* < 0.05), depression (0.63 ≤ r ≤ 0.74; *p* < 0.01), rumination (r = 0.37; *p* < 0.01), and fear of missing out (r = 0.34; *p* < 0.01) were mentioned by three studies [26,28,30].

#### 3.4.3. Stress

Seven [26,30,37,38,45,47,52] of the nine studies [26,30,37,38,40,45,47,49,52] that mentioned stress as a factor associated with PSU highlighted a positive correlation between PSU and perceived stress in 7374 participants, with a correlation coefficient ranging from 0.14 to 0.508 and degrees of significance ranging from 0.0005 to 0.5. For example, the stress prevalence in Saudi Arabian students with an average age of around 23 years was 41.26 percent, and more students with PSU belonged to the stressed group (*p* = 0.0001) [49].

Regarding the relationship between stress and sociodemographic variables, Omani male students were more stressed than females, and students living off-campus were more stressed than those living on campus [30]. According to the findings of two studies, Kuwaiti and Jordanian female students were more stressed than males [26,29]. A single study found no correlation between perceived stress and gender or academic performance [52].

In terms of smartphone use patterns, stress was significantly correlated with the duration of smartphone use (r = 0.21; *p* < 0.001) among Emirati students aged 18 to 24 years (r = 0.21; *p* < 0.001) [37].

Concerning the relationship between anxiety and other mental health disorders, stress was found to be positively correlated with insomnia (r = 0.449; *p* < 0.05) in Emirati students [37] and negatively correlated with life satisfaction (r = −0.492; *p* < 0.0005) in Lebanese students [52]. Furthermore, stress was found to be positively and significantly related to depression (r = 0.65; *p* < 0.01) and anxiety (r = 0.67; *p* < 0.01) [26].

Figure 2 depicts a summary of the current systematic review findings.

## 4. Discussion

PSU was defined by Goodman as the failure to control the use of smartphones, which adversely influences users [56]. Griffiths insists on addiction symptoms, such as salience, mood modification, tolerance, withdrawal, conflict, and relapse [57]. This disparity in characterizing the concept of PSU resulted in the use of various terms to denote excessive smartphone use, such as “addiction”, “problematic use”, or “dependence”.

The prevalence of PSU ranged from 4.3 percent to 97.8 percent in the current systematic review. This significant disparity can be explained in a variety of ways. The first is the lack of a standardized tool to assess the prevalence of PSU in all the studies included in this review; instead, 13 different validated scales (e.g., SAS, SAS-SV, MPPUS-10, MPAI, SPAI…) were used to assess the prevalence of PSU in the included studies. The second factor to consider was the population studied, which involved the general population, office workers, nurses, university students, and female secondary school students. Furthermore, various PSU thresholds were used, and several studies only reported the average score rather than the prevalence. It was extremely difficult to compare results, consequently. This systematic review is also intended to investigate the factors that influence PSU. PSU was associated with some time factors, such as average duration of daily mobile use, checking smartphone during the night, frequency of mobile phone use, and the time one spends before consulting their mobile phones when getting up in the morning, according to the findings of various studies in the MENA region. These findings were consistent with numerous studies conducted in different parts of the world [2,58,59,60]

PSU was also linked to sociodemographic factors, such as age. Several studies conducted worldwide [3] found that young people were more vulnerable to PSU, as is the case in the MENA region. Gender, marital status, income, educational level, and the type of residence of smartphone users produced mixed results. There was no general tendency in the relationship between PSU and the previously stated variables, as had been demonstrated in previous research in other parts of the world [60].

PSU, depression, anxiety, and stress were all interconnected in terms of mental health. However, it is worth noting that the association between PSU and depression was stronger than the association between anxiety and stress. These findings are consistent with other studies conducted outside the MENA region [3,4,7].

Females and younger people are more prone to depression [61,62,63]. Numerous studies have been proposed to clarify this mood disorder, the risks of which may be social, educational, or biological in nature. Females and younger people are more prone to depression [63], motivating them to engage in virtual relationships via smartphone applications, such as social networks, which increases the frequency and duration of their device consultation, resulting in PSU. Furthermore, females are predisposed to depression due to cyclical variations in sex hormone concentration [64,65]. Since many studies have been conducted on students, it is assumed that depression is caused by the stressful environment of schools or universities, as well as the lack of space and recreation time. To distract themselves, students frequently seek refuge in smartphone functions, such as games and interpersonal communication, which can be addictive [66,67].

Some evidence in the study suggested that PSU in the MENA region was linked to subsequent stress, anxiety, and insomnia, which was supported by other studies [68,69,70]. The association between PSU and insomnia could be explained by the fact that excessive smartphone use causes the user to stay up late at night to not miss a message, information, or stage of an online game, which causes sleep disruption due to exposure to radiation and blue light from smartphone screens, which affects the onset time of melatonin [68,71].

Regarding the association between PSU and mental health in the MENA region, studies evoked that self-esteem and satisfaction with life were correlated with depression and stress among smartphone users. Thus, PSU can be considered a social issue caused by a lack of face-to-face social relationships [72]. This weak engagement in social activities among smartphone users promotes unhealthy interpersonal trust in the individuals, which influences their self-esteem [73] and life satisfaction in general [74,75].

To measure the prevalence of PSU and mental health outcomes, researchers usually used a self-report questionnaire which is considered a subjective way of collecting data. However, a single study objectively and subjectively measured the use of smartphones at night [36] and demonstrated that there was a difference between the two measurements and that the objective measure most closely matched reality. Based on the findings of this study, it will be exceptionally beneficial to further investigate PSU and its associated mental disorder by using objective measurements for smartphone exposure and formal diagnosis for mental health problems. The coupling of data collection based on subjective methods (self-administered questionnaires) and those resulting from objective methods (diagnostics and objective measurements) is not taken for granted, which is why there is an interest in seeking opportunities for adequate funding for the subject of study that will contribute to improving the quality of the data obtained. Furthermore, an improvement in the quality of studies in this field during statistical analyses is required to have more convincing results concerning the relationship between PSU and mental health distress and not to be content with simple correlational studies. Without forgetting to emphasize the importance of cohort studies in confirming the causal relationships between variables in a study, there is a pressing need to conduct this type of study on this subject to highlight the impact of PSU on individuals’ mental health.

Another intriguing aspect of this review was the inclusion of a study [22] that focused on the role of metacognition in PSU and found that smartphone metacognitions predicted PSU. Metacognition is one of the psychological approaches that explain addictive behavior emergence. Wells conceptualized it as a conductor of thought. Indeed, the metacognitive model of addictive behavior confirms that metacognitive beliefs play a critical role in the generation and persistence of behaviors [76]. Several studies have found a significant positive relationship between metacognition beliefs and addictive behaviors [77,78], supporting the efficacy of interventions that target metacognition or metacognitive therapy for people suffering from PSU.

## 5. Conclusions

In conclusion, this systematic review summarized the findings of all English-language studies that looked at the relationship between PSU and mental health, specifically depression, anxiety, and stress, in MENA countries. Unfortunately, the cross-sectional design of most studies included in this review hampered determining the causal relationship between PSU and mental health problems. However, we can still conclude that PSU is a real public health issue related to mental illnesses and social dysfunctions, which constitute a wearisome burden for society. Given that the management of mental illnesses in Morocco, as in other MENA countries, suffers from a lack of specialized human resources and reception structures (e.g., 557 psychiatrists and psychologists for a population of 37.08 million in Morocco) [79], there is an urgent need to implement preventive measures, particularly at the school level, because young people are the most affected. These preventive measures must be tailored to the cultural context and socioeconomic needs of the region’s population.

## Figures and Tables

**Figure 1 ijerph-20-02891-f001:**
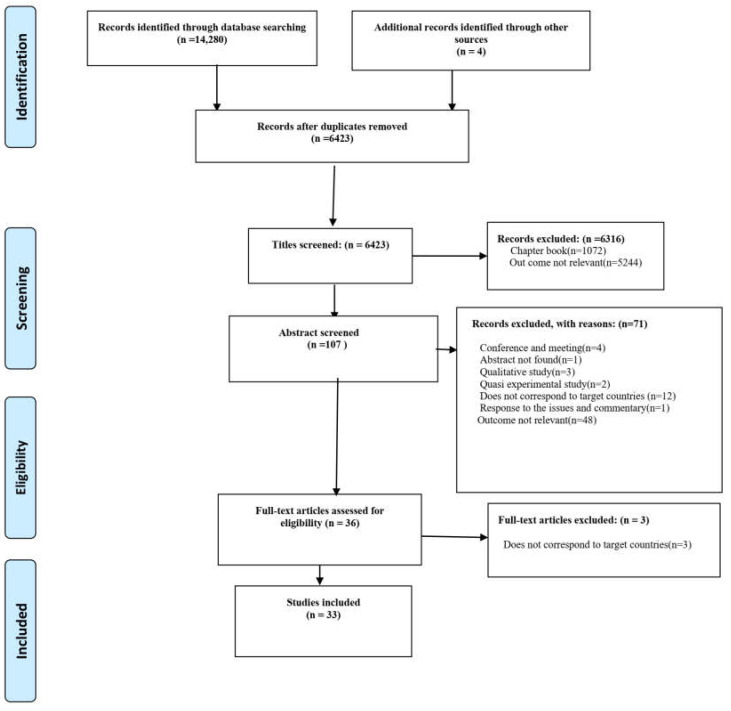
Flow diagram of the process of systematic literature search in accordance with PRISMA guidelines.

**Figure 2 ijerph-20-02891-f002:**
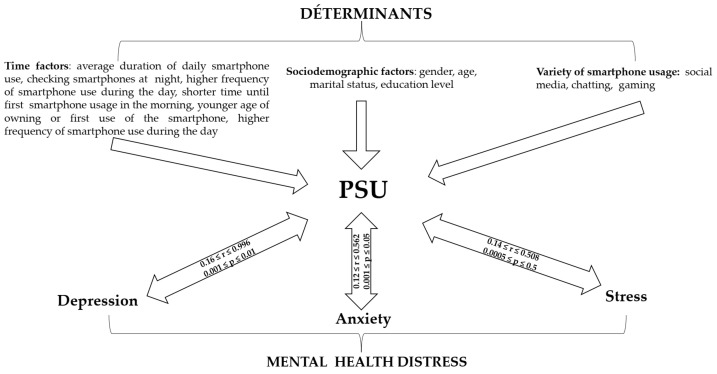
Relationships between PSU, its determinants, and mental health. r/Correlation coefficient and *p*/*p*-value.

**Table 1 ijerph-20-02891-t001:** Main results of included studies.

Source	Year of Study	Study Design	Country	Sample Size	Age Range (Mean Age ± SD)	% Female	Definitions	Mean Score or Prevalence	Assessment Tool of PSU	Assessment Tool of Depression, Anxiety, and Stress	Outcomes
Akbari M. et al., 2021 [22]	Not reported	Cross-sectional	Iran	618	15–67 (27.31 ± 8.95)	63.6	PSU shares abstinence symptoms with substance and behavioral addictions	31.51 ± 10.37	SAS-SV	HADS	Positive correlation of anxiety and depression with PSU (*p* < 0.01), and depression predicted PSU level
Okasha T. et al., 2021 [23]	2019–2020	Cross-sectional	Egypt	1380	18–26 (20.525 ± 1.576)	55	His inability to regulate his smartphone use affects other aspects of his life	59.57% (38.07 ± 12.95)	SAS-SV	BDI BAI	High significant correlation between PSU, depression, and anxiety
Barzegari S. et al., 2021 [24]	Not reported	Cross-sectional	Iran	281	18–39 (20.9 ± 2.57)	55.2	Excessive use of smartphones disrupts the daily life of users	55.86 ± 14.17	SPAI	PHQ-9	Positive correlation between PSU and depression (r = 0.47; *p* < 0.001)
Zeidan J. et al., 2021 [25]	2020	Cross-sectional	Lebanon	461	(22.25 ± 2.87)	70.9	An inability to regulate one’s use of the smartphone, which creates problems with social and psychological levels	31.19 ± 8.80	SAS-SV	TEMPS-M	PSU associated with depression (r = 0.358) and anxiety (r = 0.27) (*p* < 0.001)
Buabbas A.J. et al., 2021 [26]	Not reported	Cross-sectional	Kuwait	1993	11–21 (15.28 ± 1.71)	52.5	Not reported	64.6%	SAS-SV	DASS-21	A correlation between PSU, stress (r = 0.42), anxiety (r = 0.29), and depression (r = 0.32) (*p* < 0.01)
Alageel A.A. et al., 2021 [27]	Not reported	Cross-sectional	Middle East	506	≥21	68.77	Consist of compulsive behaviors, tolerance, withdrawal, and functional impairment	51%	SAS	PHQ9	Association between PSU and Major Depressive Disorder (MDD) (r = 0.408; *p* = 0.001)
Vally Z. et al., 2021 [28]	2019–2020	Cross-sectional	United Arab Emirates	261	18–36 (21.51 ± 2.99)	65.1	PSU is accompanied by functional impairment and symptoms that are observed in substance use disorders	(35.17 ± 8.67) Female (32.53 ± 7.60)Male	SAS-SV	DASS-21	PSU related to depression (r = 0.18; *p* < 0.01) and anxiety (r = 0.20; *p* < 0.01)
Sanusi S.Y. et al., 2021 [29]	2017–2018	Cross-sectional	Jordan	420	17–27 (20.9 ± 1.62)	75.5	Not reported	109.9 ± 23.83	SAS	PSS-10	A significant correlation between perceived stress and sleep quality and a significant correlation between PSU and sleep quality
Al Battashi N. et al., 2020 [30]	2019	Cross-sectional	Oman	404	18–26 (21.3 ± 1.6)	64.1	Not reported	83.9 ± 30.4	SAS	DASS	Significant positive correlation between PSU, anxiety, and stress
El-Sayed Desouky D. et al., 2020 [31]	2017–2018	Cross-sectional	Saudi Arabia	1513	(20.58 ± 1.71)	54.5	The excessive uncontrolled use of the smartphone, despite the awareness of the consequences and the presence of withdrawal symptoms	59.51 ± 16.93	PUMP	TMAS, BDI	PSU correlated with depression (r = 0.534; *p* < 0.001) and anxiety (r = 0.225; *p* < 0.001). Being female, of older age, or having depression or anxiety were risk factors for PSU
Derakhshanrad N. et al., 2020 [32]	2018–2019	Cross-sectional	Iran	1602	(42.2 ± 8.2) SNPG, (43.2 ± 8.8) APG	64.1	Not reported	20.3% (23.1% male, 18.8% female)	SAS-SV	DASS-42	PSU prevalence increases with depression, anxiety, and stress (*p* < 0.001)
Fallahtafti S. et al., 2020 [33]	Not reported	Cross-sectional	Iran	389	12–18	52	Not reported	30.85 ± 10.67	SAS-SV	KADS	Correlation between PSU and depression (r = 0.41; *p* < 0.001)
Turgeman L. et al., 2020 [34]	2019	Cross-sectional	Israel	140	22–35 (26 ± 3.38)	55.50	Excessive use despite adverse consequences, withdrawal phenomena, and tolerance	96.22 ± 33.56	SAS	LSAS	PSU is associated with high levels of social anxiety
Mohamed S.M. et al., 2020 [35]	Not reported	Cross-sectional	Egypt	320	Not reported	54.7	Form of behavioral addiction, including salience, tolerance, withdrawal symptoms, lies, interpersonal and intrapersonal conflict, and relapse	95.8%	SAS	HRSD	Significant positive correlation between PSU and depression
Shoval D. et al., 2020 [36]	2019	Cross-sectional	Israel	40	19–30 (23 ± 2.4)	100	Not reported	Not reported	O.S.S.N.I.Q.	STAI, BDI-II	Significant positive correlation between night-time smartphone use on psychological well-being (trait anxiety and depression)
Vally Z. et al., 2020 [37]	2019–2020	Cross-sectional	United Arab Emirates	453	18–24 (20.32 ± 1.53)	74.2	Inability to control smartphone use, increasing tolerance, and withdrawal symptoms	22.56 ± 5.03	SABAS	DASS-21	Positive and significant associations with depression, anxiety, and stress
Mosalanejad L. et al., 2019 [38]	2014	Cross-sectional	Iran	224	Not reported	82.14	Not reported	97.8%	S.A.Q.	DSI	Stress correlated with with PSU (r = 0.269; *p* < 0.05)
Miri M. et al., 2019 [39]	2018	Cross-sectional	Iran	353	(25.07 ± 6.29)	75.5	Not reported	72.6% MD 2.4% SD	PMPAS	SF-12, MCS	Inverse relationship between mental component and PSU (*p* < 0.001)
Saberi H. et al., 2019 [40]	2016	Cross-sectional	Iran	222	18–50 (26.8 ± 5.82)	73	Not reported	14.4%	P.Q.D.M.	S.Q.D.M.	Stress had a significant relationship with PSU (*p* = 0.003)
Ranjbaran M. et al., 2019 [41]	Not reported	Cross-sectional	Iran	334	(22.29 ± 3.50)	79	Not reported	119.83 ± 43.53	MPPUS	GHQ-28	Positive correlation between PSU and total score of GHQAnxiety is a significant predictor of PSU
Vally Z. et al., 2019 [42]	2018	Cross-sectional	United Arab Emirates	350	18–33 (20.7 ± 2.14)	74.4	Not reported	29% (47.14 ± 19.98).	MPPUS-10	CESD-10	Significant association between PSU and depression Gender and depression are significant predictors of PSU
Alhassan A.A. et al., 2018 [43]	2017	Cross-sectional	Saudi Arabia	935	≥18 (31.7 ± 11)	66.2	Preoccupation, tolerance, lack of control, withdrawal, conflict, lies, excessive use, and loss of interest	17%	SAS-SV	BDI II	A significant positive linear relationship between PSU and depression
Mahmoodi H. et al., 2018 [44]	2015	Cross-sectional	Iran	1034	13–21	63.64	Inability to regulate smartphone use, which involves negative consequences in daily life	4.3%	MPAI	GHQ	PSU increases the odds of poor mental health by 3.19 times.
Lin C.Y. et al., 2018 [45]	2017–2018	Cross-sectional	Iran	3807	(15.53 ± 1.2)	46.9	Complex and composite behavior that causes functional impairment, lack of control, and/or dysfunctional coping	18%	SABAS	DASS	Correlation between PSU and depression (r = 0.16; *p* < 0.01), anxiety (r = 0.49; *p* < 0.01), and stress (r = 0.32; *p* < 0.01)
Nahas M. et al., 2018 [46]	Not reported	Cross-sectional	Lebanon	207	18–65 (12.5% 35–64) (27% 18–34)	52.5	PSU is characterized by compulsive behavior, functional impairment, withdrawal, and tolerance	20.2%	MPPUS-10	PHQ-2	No correlation between PSU and psychological problems, such as depression
Zeeni N. et al., 2018 [47]	Not reported	Cross-sectional	Lebanon	244	16–21	63.93	Not reported	Not reported	MTUAS	DASS-21	Stress, anxiety, and depression are positively correlated with PSU
MatarBoumosleh J. et al., 2017 [48]	2014–2015	Cross-sectional	Lebanon	688	(20.64 ± 1.88)	47	PSU is accompanied by preoccupation, tolerance, craving, impairment of daily life functions, and withdrawal.	54.45 ± 15.65 male; 56.45 ± 14.26 female	SPAI	PHQ-2 GAD-2	PSU is significantly associated with depression and anxiety
Venkatesh E. et al., 2017 [49]	2016	Cross-sectional	Saudi Arabia	189	23.28 male, 23.30 female	46.56	Overuse of smartphones disturbs users’ daily lives.	71.96%	SAS-SV	I.Q.S.	High-stress levels are significantly associated with PSU
Al-Dossary N.A. et al., 2017 [50]	2016–2017	Cross-sectional	Saudi Arabia	493	>15	100	Continuous consultation with smartphones despite adverse effects, loss of self-control, compulsive participation, and cravings	58%	SAS-SV	BDI-PC	PSU and depression were significantly and positively correlated
Hawi N.S. et al., 2017 [51]	2016	Cross-sectional	Lebanon	381	17–27 (20.84 ± 1.92)	Not reported	Not reported	Not reported	SAS-SV	BAI	PSU increases the odds of having high anxiety by 4.706
Samaha M. et al., 2016 [52]	Not reported	Cross-sectional	Lebanon	249	17–26 (20.96 ± 1.93)	45.8	Not reported	44.6%	SAS-SV	PSS-10	Positive correlation (r = 0.2; *p* < 0.002) between the risk of PSU and perceived stress
Tavakolizadeh J. et al., 2014 [53]	2011	Cross-sectional	Iran	700	18–30	44	Not reported	36.7%	MPAI	GHQ-28	Significant association between PSU, mental health status (*p* = 0.001), depression, and anxiety
Babadi- Akashe Z. et al., 2014 [54]	Not reported	Cross-sectional	Iran	296	Not reported	49.90	Mental impairment resulting from modern technology	15.7%	32-P.S.Q.	SCL-90-R	Significant negative relationship between mental health and PSU (r = −0.383; *p* < 0.001).

SD/standard deviation; r/correlation coefficient; *p*/*p*-value; PSU/Problematic Smartphone Use; SAS-SV/Smartphone Addiction-Short Version; HADS/Hospital Anxiety and Depression Scale; BDI/Beck Depression Inventory; BAI/Beck Anxiety Inventory; SPAI/Smartphone Addiction Inventory; PHQ-9/Patient Health Questionnaire; TEMPS-M/Temperament Evaluation of Memphis, Pisa, Paris, and San Diego; DASS/Depression Anxiety and Stress Scale; SAS/Smartphone Addiction Scale; PSS-10/Perception of Stress Scale; PUMP/Problematic Use of Mobile Phone; TMAS/Taylor Manifest Anxiety Scale; SNPG/Symptomatic Neck Pain Group; APG/Asymptomatic Participant Group; KADS/Kutcher Adolescent Depression Scale; LSAS/Liebowitz Social Anxiety Scale; HRSD/ Hamilton Rating Scale of Depression; O.S.S.N.I.Q./Objective and Subjective Smartphone Night-Time Use Independent Questionnaire; TAI/State-Trait Anxiety Inventory; BDI-II/Beck Depression Inventory Second Edition; SABAS/Smartphone Application-Based Addiction Scale; S.A.Q./Smartphone Addiction Questionnaire; DSI/Daily Stress Inventory; PMPAS/Mobile Phone Addiction Scale; MD/Moderate Dependence; SD/Severe Dependence; SF-12/Short Form 12 Questionnaire; MCS/Mental Component Summary; P.Q.D.M./Persian Questionnaire Designed by Mazaheri; S.Q.D.M./Second part of the Questionnaire Designed by Mazaheri; MPPUS/Mobile Phone Problem Usage Scale; GHQ-28/General Health Questionnaire-28; CESD-10/Centre for Epidemiological Studies Depression Scale; MPAI/Mobile Phone Addiction Index; MTUAS/Media and Technology Usage; GAD-2/Generalized Anxiety Disorder; I.Q.S./Independent Questionnaire of Stress; BDI-PC/Beck Depression Inventory-Primary Care Version; 32-P.S.Q./32-Point Scale Questionnaire of Behavior Associated with Mobile Phone Use; and SCL-90-R/The Symptom Checklist-90-R.

**Table 2 ijerph-20-02891-t002:** Quality assessment of included cross-sectional studies using the Newcastle-Ottawa Scale.

Source	Selection				Comparability	Exposure		Subtotal Assessment			Overall
	Representativeness of Sample	Ascertainment of Exposure	Sample Size	Non-Respondents	Confounders Are Controlled for	Assessment of Outcome	Statistical Test	S& Total	C# Total	E∑ Total	
Akbari M. et al., 2021 [22]	*	**			*		*	Good	Good	Moderate	Moderate
Okasha T. et al., 2021 [23]	*	**					*	Good	Poor	Moderate	Poor
Barzegari S. et al., 2021 [24]	*	**	*				*	Good	Poor	Moderate	Poor
Zeidan J. et al., 2021 [25]	*	**	*		*		*	Good	Good	Moderate	Moderate
Buabbas A.J. et al., 2021 [26]	*	**	*				*	Good	Poor	Moderate	Poor
Alageel A.A. et al., 2021 [27]	*	**			*		*	Good	Good	Moderate	Moderate
Vally Z. et al., 2021 [28]	*	**		*			*	Good	Poor	Moderate	Poor
Sanusi S.Y. et al., 2021 [29]	*	**	*		*		*	Good	Good	Moderate	Moderate
Al Battashi N. et al., 2020 [30]	*	**	*				*	Good	Poor	Moderate	Poor
El-Sayed Desouky D. et al., 2020 [31]	*	**			*		*	Good	Good	Moderate	Moderate
Fallahtafti S. et al., 2020 [33]	*	**					*	Good	Poor	Moderate	Poor
Turgeman L. et al., 2020 [34]		**			*		*	Moderate	Good	Moderate	Moderate
Mohamed S.M. et al., 2020 [35]	*	**	*				*	Good	Poor	Moderate	Poor
Shoval D. et al., 2020 [36]		**					*	Moderate	Poor	Moderate	Poor
Vally Z. et al., 2020 [37]		**					*	Moderate	Poor	Moderate	Poor
Mosalanejad L. et al., 2019 [38]	*	**	*		*		*	Good	Good	Moderate	Moderate
Miri M. et al., 2019 [39]	*	**	*				*	Good	Poor	Moderate	Poor
Saberi H. et al., 2019 [40]		**					*	Moderate	Poor	Moderate	Poor
Ranjbaran M. et al., 2019 [41]	*	**			*		*	Good	Good	Moderate	Moderate
Vally Z. et al., 2019 [42]		**			*		*	Moderate	Good	Moderate	Moderate
Alhassan A. et al., 2018 [43]	*	**			*		*	Good	Good	Moderate	Moderate
Mahmoodi H. et al., 2018 [44]	*	**	*		*		*	Good	Good	Moderate	Moderate
Lin C.Y. et al., 2018 [45]	*	**					*	Good	Poor	Moderate	Poor
Nahas M. et al., 2018 [46]	*	**	*				*	Good	Poor	Moderate	Poor
Zeeni N. et al., 2018 [47]		**					*	Moderate	Poor	Moderate	Poor
Matar Boumosleh J. et al., 2017 [48]	*	**	*		*		*	Good	Good	Moderate	Moderate
Venkatesh E. et al., 2017 [49]		**					*	Moderate	Poor	Moderate	Poor
Al-Dossary N.A. et al., 2017 [50]	*	**	*				*	Good	Poor	Moderate	Poor
Hawi N.S. et al., 2017 [51]	*	**			*		*	Good	Good	Moderate	Moderate
Samaha M. et al., 2016 [52]	*	**			*		*	Good	Good	Moderate	Moderate
Tavakolizadeh J. et al., 2014 [53]	*	**					*	Good	Poor	Moderate	Poor
Babadi-Akashe Z. et al., 2014 [54]	*****	******					*	Good	Poor	Moderate	Poor

*–Study adequately filled criteria for this sub-domain; **–When the study uses valid instruments it is offered 2 stars for ascertainment of the exposure in the selection domain; S&–selection total; C#–comparability total; and E∑–exposure total.

**Table 3 ijerph-20-02891-t003:** Quality assessment of included cohort studies using the Newcastle-Ottawa Scale.

Source	Selection				Comparability	Exposure			Subtotal Assessment			Overall
	Representativeness of Sample	Ascertainment of Exposure	Sample Size	Demonstration That Outcome Aas Not Present at the Beginning of Study	Confounders Are Controlled for	Assessment of Outcome	Length of Follow-Up	Follow-Up Rate	S& Total	C# Total	E∑ Total	
Cohort Studies												
Derakhshanrad N. et al., 2020 [32]	*****	******		*****	*****		*****		Good	Good	Moderate	Moderate

*–Study adequately filled criteria for this sub-domain; **–When the study uses valid instruments it is offered 2 stars for ascertainment of the exposure in the selection domain; S&–selection total; C#–comparability total; and E∑–exposure total.

**Table 4 ijerph-20-02891-t004:** Thresholds for Quality Assessment using the Newcastle-Ottawa Scale [55].

Quality Rating	Points in Selection Domain	Points in Comparability Domain	Points in Exposure Domain
Good	≥3	≥2	≥2
Moderate	2	≥1	≥2
Poor	0–1	0	0–1

## Data Availability

Not applicable.

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
