# Peer review of "Association between Problematic Use of Smartphones and Mental Health in the Middle East and North Africa (MENA) Region: A Systematic Review"

_ijerph, 2023, doi:10.3390/ijerph20042891_

Round 1

Reviewer 1 Report

The manuscript (Association between problematic use of smartphones and mental health in the Middle East and North Africa (MENA) region: a systematic review)  examines the findings of studies that looked at the relationship between PSU and mental health, with particular reference to the depression, anxiety and stress in MENA countries.

The manuscript makes a significant contribution to this field of study. However there are some critical points:

1 the review in the title considers a geographical reference that includes the Middle East and North Africa (MENA). However, no works are reported that analyze the situation in the regions of North Africa with the exception of two works relating to Egypt (Okasha et al. 2021; Mohamed et al., 2020).  A third work makes a generic reference to the Middle East. The country should be specified as for the others.

 For these reasons, perhaps it would be appropriate to reduce the review to the countries of the Middle East and avoid considering Algeria, Bahrain, Djibouti, Ethiopia, Iraq, Mauritania, Morocco, Libya, Palestine, Qatar, Sudan, Syria, Tunisia, or Yemen. By the way, some of these countries are not North Africa.

 2. Table 1

Table 1 should be reviewed, presented in this way it is difficult to read. A simplification and sybthesis of the content would be necessary.

The bibliographic reference (Column 1) could dispense with the name and use the usual method of references. Es Akbari M. et al. 2021.

 Column “Definition of PSU” should become “definitions”since the definitions are different; for this reason they should, if possible, be further summarized.

Outcomes column: present more succinctly

In general, in the discussion it would have been appropriate to underline the critical aspects of the works examined.

Reviewer 2 Report

An authors’ preprinted version was found on https://assets.researchsquare.com/files/rs-1662848/v1/88a90938-fd4b-4914-af12-8d9a77b98462.pdf?c=1662112217

Thank you for allowing me to review a decent work of yours. It is an interesting topic. The paper has quite a comprehensive review of the problematic use of smartphones.

The method is fitting to the objective. However, The argument for the purpose is not strong enough. Why this region would be of interest to the audience of the journal? They have to argue why they have chosen study regions instead of a general review.

In the same sense, they should have indicated clearly what would be the specific implications in the conclusions, for the reason for the region that I have chosen.

The review of the articles is reasonably good. But the tables are ugly and formatted, which takes up too much space. The authors should make adjustments to that.

As the authors conclude, there is a big disappointment in determining the causal relationship among the entities under study. From session 3, obviously, there is a hierarchical relationship among the constructs of interest. For the presentation and for the highlights of the contribution of the research, it would be good to produce a visual aid of the charts.

Actually, somehow I disagree with this conclusion (of the absence of a causal relationship) because in the review, many articles clearly show the correlations of the constructs. And within those articles, hypotheses had argued the causal relationships theoretically. Thus, right from the original articles, they could have recreated the casual relationships among the constructs. If the authors choose to do that, they would have built up a large network of causal relations, which could be quite eye-catching in a diagram form.

As a whole, the review work is good; but the paper is not complete. The arguments in the introduction and the implications need to be reinforced so that it can look more organized, and show a more significant contribution.
